# Understanding the Role of Perilipin 5 in Non-Alcoholic Fatty Liver Disease and Its Role in Hepatocellular Carcinoma: A Review of Novel Insights

**DOI:** 10.3390/ijms22105284

**Published:** 2021-05-17

**Authors:** Paola Berenice Mass Sanchez, Marinela Krizanac, Ralf Weiskirchen, Anastasia Asimakopoulos

**Affiliations:** Institute of Molecular Pathobiochemistry, Experimental Gene Therapy and Clinical Chemistry (IFMPEGKC), RWTH University Hospital Aachen, D-52074 Aachen, Germany; pmasssanchez@ukaachen.de (P.B.M.S.); mkrizanac@ukaachen.de (M.K.)

**Keywords:** perilipin 5, non-alcoholic liver disease, fatty liver, hepatocellular carcinoma, cancer

## Abstract

Consumption of high-calorie foods, such as diets rich in fats, is an important factor leading to the development of steatohepatitis. Several studies have suggested how lipid accumulation creates a lipotoxic microenvironment for cells, leading cells to deregulate their transcriptional and translational activity. This deregulation induces the development of liver diseases such as non-alcoholic fatty liver disease (NAFLD) and subsequently also the appearance of hepatocellular carcinoma (HCC) which is one of the deadliest types of cancers worldwide. Understanding its pathology and studying new biomarkers with better specificity in predicting disease prognosis can help in the personalized treatment of the disease. In this setting, understanding the link between NAFLD and HCC progression, the differentiation of each stage in between as well as the mechanisms underlying this process, are vital for development of new treatments and in exploring new therapeutic targets. Perilipins are a family of five closely related proteins expressed on the surface of lipid droplets (LD) in several tissues acting in several pathways involved in lipid metabolism. Recent studies have shown that *Plin5* depletion acts protectively in the pathogenesis of liver injury underpinning the importance of pathways associated with PLIN5. PLIN5 expression is involved in pro-inflammatory cytokine regulation and mitochondrial damage, as well as endoplasmic reticulum (ER) stress, making it critical target of the NAFLD-HCC studies. The aim of this review is to dissect the recent findings and functions of PLIN5 in lipid metabolism, metabolic disorders, and NAFLD as well as the progression of NAFLD to HCC.

## 1. Introduction

Lipid accumulation in the liver plays a pivotal role in the pathogenesis of NAFLD that might result from dysfunction of cellular lipid trafficking and lipid droplet formation. The accepted model of LD formation is based on the biosynthesis of neutral lipids (e.g., triacylglycerides, TAGs) by esterification of fatty acids (FA) into diacylglycerol or sterols [1,2]. The enzymes involved in this biosynthesis of sterols are acetyl-CoA acetyltransferases 1 and 2 (ACAT1 and ACAT2), whereas TAGs are the product of diacylglycerol acyltransferases (DGAT1 and DGAT2). Once TAG accumulates in the ER, a vesicle of lipids is formed in the ER membrane and LDs are released. Typically, the LD surface is covered with a monolayer of phospholipids and proteins [1,2]. Several enzymes are located in ER, helping the formation, stabilization, and degradation of LDs. These proteins belong to the perilipin/PAT family, the Ras superfamily of GTPases (Rab), and ADP-ribosylation factor1-coat protein complex I (Arf-COPI) [3]. In this context, the general structure of LD consists of a central core of TAG and cholesterol esters, while a peripheral monolayer is composed of phospholipids with attached proteins with structural and functional roles as depicted in Figure 1 [1].

Perilipins were initially localized in the phospholipid monolayer of LDs in the 1990s [4,5,6] and it was not until 2010 that the “perilipin family protein: PAT protein” was suggested to unify the name of this family [7]. Perilipins are a family comprising five distinct proteins, whose expression varies depending on the tissue. PLIN1 presents its major expression in white adipocyte tissue (WAT). It can even be expressed in brown adipocyte tissue (BAT) or cardiac muscle and its functions include hormone-induced lipolysis and large LD stabilization [8,9]. PLIN2 is mainly expressed in liver and pre-adipocytes and is involved in adipocyte differentiation, small LD formation and stabilization [9,10]. PLIN3 expression is ubiquitous and the protein functions as a LD stabilizer and is necessary for intracellular lipid trafficking [9]. The fourth member, PLIN4, is expressed in WAT and skeletal muscle and is involved in human adipocyte differentiation [9,11]. PLIN5 is expressed in oxidative tissues such as cardiac muscle, BAT, skeletal muscle, and liver and is functionally linked to LD stabilization and FA supply to mitochondria [9].

NAFLD is characterized by augmented accumulation of hepatic LD, which is in line with increased PLIN5 expression [12]. Recently, alterations in lipid metabolism have also been proposed as a new hallmark of cancer [13,14,15]. Moreover, several cellular processes such as oxidative stress, ER stress, and inflammation have been related to non-alcoholic steatohepatitis (NASH) the progressed form of NAFLD [16,17] as well as to cancer development and progression [18,19]. Meanwhile, PLIN5 has indeed been implicated in the regulation of lipid and glucose homeostasis and many cellular processes such as oxidative stress, ER stress, inflammation, and autophagy [12,20,21,22,23,24,25,26,27,28,29,30]. This evidence once again supports the notion that PLIN5 could play a direct role in transition from NAFLD to HCC. Therefore, it was suggested that knowing precisely how the regulation of PLIN5 in the NAFLD-HCC pathogenesis occurs, will potentially help to identify new targets for the diagnosis and prognosis of respective pathologies.

## 2. An Overview of *Plin5* Transcriptional Regulation and Interactions

In humans and mice, PLIN5 is a protein consisting of 463 residues. In humans, *PLIN5* maps to chromosome 19p13.3 and in mice to chromosome 17. The gene is composed of eight exons in humans and of nine exons in mice, while the loci span approximately 12.9 kbp and 5.9 kbp, respectively, both encoding transcripts of 1.9 kb in size [31]. PLIN5 was independently discovered by three groups [31,32,33]. In one study, PLIN5 was named myocardial LD protein (MLDP) due to its cardial expression [32]. The other two groups reported the expression of PLIN5 in oxidative tissues-enriched PAT protein (OXPAT) and lipid storage droplet protein 5 (LSDP5), a protein not exclusively expressed in the heart, but highly expressed in oxidative tissues, such as muscle and liver [31,33]. Peroxisome proliferator-activated receptor (PPAR) was shown to regulate the transcription of *Plin5* since the first intron of the gene contains a PPAR regulatory element (PPRE) [31]. All three groups showed that *Plin5* is induced in the liver under fasting conditions in a PPARα-dependent fashion [31,32,33]. Furthermore, the expression of PLIN5 was found to be induced by PPARα agonists in liver, skeletal, and cardiac muscle and in WAT [34]. In line with this, a study performed in porcine kidneys showed that the transcriptional factor CCAAT/enhancer-binding protein alpha (C/EBPα) binds to the promoter region of *Plin5*, inducing its expression under the fasting state [35]. It has been also reported that Jun proto-oncogene (JUN), activating transcription factor (ATF)1, ATF3, and ATF4 can bind to the *Plin5* promoter and induce its expression in hepatocytes [26].

Different enzymes, such as adipose triglyceride lipase (ATGL), hormone-sensitive lipase (HSL), and monoglyceride lipase (MGL) regulate lipid hydrolysis. TAG hydrolysis is catalyzed by ATGL, followed by cleavage of one molecule of FA forming diaglycerides by HSL and finally hydrolysis is completed by MGL [1]. The interaction between PLIN5 and ATGL and its protein activator, α-β-hydrolase domain-containing 5 (ABHD5), results in decreased lipolysis [36,37]. In this scenario, different lipid droplet proteins play a key role in regulating lipid metabolism and TAG storage in cells.

These findings elucidate the participation of PLIN5 in lipid metabolism. Several studies have tested different approaches in order to have a better understanding of how PLIN5 exerts its functions in a cell-specific manner. Under this premise, the question arises of how perilipins, specifically PLIN5, are involved in disorders such as NAFLD in which LD formation is deregulated.

## 3. Understanding NAFLD and Its Progression to HCC

The progression of NAFLD towards the development of HCC and how risk factors can promote its appearance is well documented [38,39]. Historically, NAFLD has been described as a spectrum of chronic liver diseases characterized by hepatic lipid accumulation as well as excess of TAG in the cytoplasm [40]. Diverse factors are involved in the appearance of these pathologies. These include the excessive caloric intake that potentially leads to obesity and diabetes as well as the consumption of foods rich in fats and sucrose accompanying a sedentary lifestyle. Moreover, genetic modifications such as single-nucleotide polymorphisms (SNPs), epigenetic modifications, and viral infections (e.g., hepatitis C) are further critical risk factors for the development of HCC [41,42,43,44].

A “two-hit theory” has been proposed to describe the NASH pathogenesis. The first hit is steatosis, the accumulation of fat in the liver, followed by the second hit which is increased susceptibility of hepatocytes to secondary injuries such as oxidative stress, mitochondrial dysfunction, and release of pro-inflammatory cytokines [45,46,47,48,49]. Figure 2 illustrates the main events that take place during progress of NAFLD to NASH and eventually HCC.

According to histopathological features, steatosis is characterized by increased storage of TAGs in hepatocytes [40]. A remarkable characteristic of this stage is the ballooning of hepatocytes, which is currently considered as a hallmark of NAFLD [40]. Steatosis is often associated with increased lipolysis and insulin resistance resulting in elevation of serum free fatty acid (FFA) levels, followed by TAG relocation from the liver to peripheral organs. When inflammation and hepatocyte injury occur, the pathogenesis shifts to NASH [40,50]. In this scenario, a source of free radicals, induced by TAG accumulation, is capable of inducing oxidative stress, initiating lipid peroxidation accompanied by the release of pro-inflammatory cytokines, mitochondrial damage, and ER stress aggravating the condition of NAFLD [51]. Once hepatic fibrosis and cirrhosis occur, NASH can lead to the development of HCC, which is the most severe manifestation of hepatic injury [40,50]. Fibrosis is characterized by the deposition of high density extracellular matrix (ECM) proteins in diseased tissue, formation of regenerative nodules, and impaired wound-healing response, thus leading to cirrhosis [40,50,52]. While this “two-hit” approach allows us to understand how NAFLD could progress, a more accurate model has been suggested in recent years. This “multiple parallel hits” theory suggests that not only steatosis, insulin resistance and oxidative stress can lead to NASH, but instead multiple factors including primary inflammation inducing secondary steatosis, changes in gut microbiota, and the release of serum cytokines may act in concert to contribute to different disease stages [53,54].

Currently the “gold standard” for the identification of NAFLD is histological staining of liver biopsies, which is an invasive method accompanied with further risks for the health of the patient. Unfortunately, the lack of specific biomarkers able to differentiate the progress stages of the disease presently makes proper diagnosis and treatment uncertain [51,52]. Moreover, nowadays there is not an approved drug available for the treatment of NAFLD and NASH. Presently, intensive lifestyle changes which take into account a balanced, low-caloric diet and increased physical activity are the general recommendation to avoid disease progression [55,56]. Therapy for patients with metabolic syndrome and HCC, is focused on defeating cancer. In this regard, there is still a long way to go to identify new therapeutically effective drugs or specific biomarkers helpful in management of this complex pathology [52].

## 4. The Role of PLIN5 in Lipid Metabolism and NAFLD

As mentioned before, NAFLD is characterized by increased accumulation of LD in the liver as well as increased expression of PLIN5 [12]. This has been previously reported in patients with severe NAFLD as well as in mice fed with a high fat diet (HFD) inducing NAFLD [12,23]. Interestingly, compared to PLIN5 protein levels, mRNA levels are only slightly increased, suggesting that elevated PLIN5 activity is regulated post-transcriptionally [12].

One of the first studies to elucidate the role of PLIN5 in hepatocytes demonstrated that PLIN5 expression is induced by FFA in a time-independent manner [57]. Additionally, a study performed in *Plin5*-deficient mouse hepatocyte AML-12 cells showed a decrease in cellular TAG content as well as an increase of mitochondrial oxidation [58]. To gain a better understanding of consequences resulting from *Plin5* depletion in vivo, several knockout (KO) models in mice have been developed. Table 1 contains key points of the major models reported as well as the feeding conditions evaluated in each of these, underpinning the notion that the role of PLIN5 in metabolism seems to be tissue-specific.

Mouse models based on HFD and *Plin5* disruption (*Plin5*^−/−^) help to understand the role of PLIN5 in steatohepatitis and beyond the phenotype observed, they have provided critical information about molecular signaling and regulation of this protein. For instance, research in *Plin5*^−/−^ mice showed an improvement of glucose tolerance in liver and insulin resistance in muscle [20]. This can be explained by enhanced lipolysis derived from FA, turning into sphingolipids, and consequently causing insulin resistance. In this scenario, *Plin5*^−/−^ mice present a phenotype characterized by insulin resistance due to impairment in glucose disposal in skeletal muscle and WAT, but not in the liver [20]. In line with this, specific hepatic knockdown of Notch1 in mice resulted in an upregulated expression of glucose-6-phosphatase (G6P) and *Plin5*, leading to a predisposition of a diabetic phenotype through development of insulin resistance [62]. These reports indicate the possible cross-talk between these genes in glucose response.

Another HFD model revealed a decrease in hepatic TAG content and a decrease in expression of enzymes related to FA synthesis in *Plin5*^−/−^ mice. Expression of pro-inflammatory and inflammatory biomarkers was increased in *Plin5*^−/−^ mice and enhanced under a HFD. ER stress-induced genes, hepatic injury markers, and mitochondrial markers were upregulated in *Plin5*^−/−^ animals, indicating the importance of PLIN5 in maintaining proper functions of hepatic cells [23]. Similarly, a study conducted in hepatocyte-specific *Plin5*^−/−^ mice revealed a decrease in TAG secretion and impairment in glucose metabolism that led to systemic insulin resistance. In the same study, *Plin5* ablation activated the c-Jun N-terminal kinase (JNK) signaling pathway and modulated insulin response [21].

A steatosis-like model of primary mouse and human hepatocytes, showed that statins decrease the levels of PLIN5, but not other LD-associated genes via the sterol regulatory element-binding protein 2 (SREBP2) [63]. Statins are commonly used in patients with hypercholesterolemia to reduce hepatic TAG content, but their mode of action is yet unknown. However, this finding provides a possible explanation on how the effects of PLIN5 regulation mimic the use of statins as a therapeutic drug during steatosis and shows the importance of PLIN5 as a possible target in this pathology.

Recently, it was reported that LD formation after stimulation with oleic acid (OA) leads to activation of PLIN5 in human liver cancer HepG2 cells via the PI3K/PPARα pathway [64]. In the respective study, it was demonstrated that the PPARα inhibitor GW6471 decreased the levels of PLIN5 in a dose-dependent manner. Moreover, the phosphatidylinositol-3-kinase (PI3K) inhibitor LY294002 diminished the levels of *PLIN5* mRNA in HepG2 cells. Thus the combination of OA treatment and inhibition of PI3K or PPARα activities showed a decrease in LDs, revealing the partial role of PLIN5 in the control of LD formation [64]. In this context, the PI3K pathway is also involved in ER stress. Insulin’s protective role during lipotoxicity induced by saturated fatty acids activated the PI3K/Akt/p53 pathway and ameliorated ER stress [65]. This evidence links PLIN5 as a possible mediator in response to ER stress.

The role of PLIN5 was also investigated in a high-glucose-induced podocyte injury model [30]. Feng and colleagues demonstrated the interplay among Akt/GSK-3β/Nrf2 and PLIN5 to reduce apoptosis, reactive oxygen species (ROS) production, and inflammatory response. Overexpression of PLIN5 alleviated the release of pro-inflammatory markers such as tumor necrosis factor-α (TNF-α), interleukin-6 (IL-6), and interleukin-1β (IL-1β) and upregulated the expression of nuclear factor erythroid 2–related factor 2 (Nrf2) and its targeted heme oxygenase-1 (HO-1) and NAD(P)H quinone dehydrogenase 1 (NQO1). In addition, protein kinase B (AKT) phosphorylating its target, glycogen synthase kinase 3-β (GSK-3β), led to Nrf2/ARE inactivation. In line with this study, PLIN5 was involved in oxidative stress via AMPK/Nrf2 [24]. *Plin5* expression was suppressed under HFD in mouse hepatic stellate cells (HSC) which surprisingly led to HSC activation. In chronic liver damage, HSC are known as the key drivers of liver fibrosis as their activation and transdifferentiation to myofibroblasts leads to deposition of collagen and extracellular matrix in the tissue. Exogenous overexpression of PLIN5 restored the formation of LDs and induced 5′ AMP-activated protein kinase (AMPK) expression leading to increase of endogenous PLIN5 and to the inhibition of HSC activation. The induction of PLIN5 via AMPK induced the transcription activity of Nrf2 and led to reduced ROS and increased levels of glutathione, indicating how PLIN5 attenuates oxidative stress in HSC.

Interestingly, emerging evidence suggests that PLIN5 can act as a co-regulator during lipolysis stimulated by catecholamine. The results indicate that phosphorylation of PLIN5 by protein kinase A (PKA) triggers PLIN5 nuclear translocation during fasting and lipolysis conditions when PLIN5 binds to the promoter region of peroxisome proliferator-activated receptor gamma coactivator 1-alpha (PGC-1α) [66]. Glucagon, a PKA activator, stimulated the phosphorylation of PLIN5 and inhibited its interaction with CGI-58 promoting lipolysis and reduced TG accumulation in the liver [67].

PLIN5 also regulates the complex sirtuin 1-deleted in breast cancer-1 (SIRT1-DBC1), favoring the deacetylase activity of SIRT1 to its target PGC-1α, leading to transcription of their targets, and promoting FA oxidation and mitochondrial efficiency [66,68]. Moreover, hepatic PKA activity is increased under a chronic HFD, while interestingly this effect was enhanced in female mice, suggesting that the hormonal status plays a pivotal role in its regulation [69]. Nevertheless, it has been shown that phosphorylation of PLIN5 at serine 155 (S155) has important effects in hepatic lipid metabolism and glycemic control by allowing PLIN5 interaction with the lipase ATGL regulating further lipolysis [36]. Furthermore, it was recently shown that the loss of S155 phosphorylation of PLIN5 in liver impacts whole-body glucose tolerance and decreases levels of FA oxidation derived from TAGs [70].

Similarly, PLIN5 plays a role in autophagy by diminishing inflammation through deacetylase activity of SIRT1 in hepatocytes under fasting conditions. SIRT1 deacetylates nuclear factor-kappa B (NF-κB), thus inflammation is attenuated by the decreased levels of target genes of NF-κB such as *Il-6*, *Il-1β*, and *Tnf-α* [71]. Furthermore, PLIN5 acts as a scaffold in chaperone-mediated autophagy in HepG2 cells demonstrating that co-localization of heat shock protein 70 (Hsp70) and PLIN5 serves as a substrate allowing degradation of LD in cells under FA stimuli, which mimics NAFLD [12]. The SIRT1–PLIN5 axis has also been studied in vitro and in vivo as a target for glycycoumarin (GCM), a compound effective against liver diseases. The respective report showed that, GCM treatment increased SIRT1 expression in palmitic acid-derived lipotoxicity in AML-12 cells, which in consequence led to decreased expression of inflammatory genes such as *Tnf-α* and *Il*-6. Thus depletion of *Plin5* or *Sirt1* decreases GCM effectivity, revealing the coupled effect of these proteins to balance inflammatory response and lipoapoptosis [71].

## 5. PLIN5 Is Responsive to Cellular Processes Altered in NAFLD

NAFLD is a condition characterized of dyslipidemia, increased oxidative stress, ROS production, and mitochondrial alterations provoking inflammatory responses [17]. There are several reports analyzing how PLIN5 participates in each of these processes. In one study, the depletion of *Plin5* in an in vivo mouse model under fasting conditions was associated with decreased LD formation and TAG content in the heart [59]. The same effect was observed in soleus muscle, BAT, and WAT. Additionally, in the heart, PLIN5 induces ROS levels while it blocks the activity of lipase ATGL suggesting that this mechanism maintains cardial LDs. In the absence of *Plin5*, FA are oxidized in mitochondria in the heart and not retained as TAG, which explains the decreased levels of TAG in this tissue [59].

In muscle, ER stress, as well as inflammation or oxidative stress, was not observed in mice lacking *Plin5* [20]. This is opposite to the liver, in which several studies have found a protective role of *Plin5* deletion in this tissue [21,23,27]. The mechanisms that regulate these tissue-specific functions remain unclear, but a few studies linked fibroblast growth factor 21 (Fgf21) as a critical mediator of the effects observed in skeletal muscle. Overexpression of *Plin5* in skeletal muscle in mice protected the liver from inflammation during HFD feeding [61]. Interestingly, PLIN5 overexpression in skeletal muscle was associated with reduced hepatic expression of inflammatory markers such as IL-6, TNF-α, and monocyte chemoattractant protein-1 (MCP1) [61]. This model also suggested that PLIN5 drives FGF21 expression in muscle, which partially is metabolically protective [61]. On the contrary, *Plin5*^−/−^ skeletal muscles showed decreased levels of FGF21 in liver and muscle tissues, leading to lower levels of the secreted protein in plasma and reduced ER stress [25]. FGF21 is a stress-inducible hepatokine, which regulates energy balance and glucose and lipid homeostasis and regulates ER stress via ATF4-CHOP [72,73]. This evidence suggests an important cross-talk among tissues as well cell-autonomous regulation, which balances metabolic responses.

Another study reported that overexpression of PLIN5 in the heart induces changes in the transcriptome of cells, especially in the signaling pathways related to mitochondrial bioenergetic processes, lipid metabolism, and cytoskeleton reorganization where interestingly, overexpression of PLIN5 led to an augment of *Nrf2* targets genes [22]. This suggests that PLIN5 overexpression triggers oxidative stress and induces upregulation of the NRF2 pathway, possibly as a compensatory mechanism against LD formation in heart [22].

In the same line, recent evidence indicates that PLIN5 can modulate oxidative stress in pancreatic β-cells via Nrf2 [29]. In a palmitate-triggered lipotoxicity model in INS-1 pancreatic β-cells, *Plin5* activated the Nrf2-ARE system by stimulating PI3K/Akt and ERK signaling pathways, while phosphorylation of JNK and p38 were decreased, indicating that these pathways are involved in the *Plin5*-mediated oxidative responses [29]. Again, this study points to a protective role of PLIN5 against oxidative stress.

Experiments in HepG2 cells showed that PLIN5 overexpression exerts a protective role during lipopolysaccharide (LPS)- or H_2_O_2_-induced oxidative stress by increasing cellular LD content and promoting contact between LD and mitochondria [26]. In this study, PLIN5 reduced ROS levels and augmented the expression of key genes in mitochondrial function including cytochrome C oxidase subunit 2 and 4 (COX2 and COX4) and citrate synthase (CS), which regulate the mitochondrial respiratory chain as well as glutathione peroxidase 2 (GPX2) and catalase (CAT), genes related to antioxidant response [26]. PLIN5 expression was also upregulated by the JNK-p38-ATF pathway, potentially by binding of JUN, ATF1, ATF3, and ATF4 to specific binding elements within the *PLIN5* promoter [26]. It has been previously reported that this pathway is involved in the response of ROS activation [74,75]. Therefore, PLIN5 can potentially act as a co-regulator in oxidative response.

In livers of *Plin5*^−/−^ mice, induction of inflammation via LPS failed to trigger expression of proteins such as NF-κB, NLR family pyrin domain-containing 3 (NLRP3), and the inflammatory protein lipocalin 2 (LCN2), indicating that PLIN5 acts as a vital protein protecting the liver from injury [27]. In this study, mitochondria of *Plin5*^−/−^ mice presented a larger size and a closer localization to the ER in an HFD model as well as increased levels of mitochondrial markers such as OPA1 mitochondrial dynamin-like GTPase (OPA1), sigma 1 receptor (SIG1R), and voltage-dependent anion-selective channel 1 (VDAC1) involved in mitochondrial structure formation and trafficking. Moreover, increased levels of mitofusin 1 (MFN1) and mitofusin 2 (MFN2) were observed, genes which participate in mitochondrial fusion, possibly explaining the observed enlargement of these organelles [27]. This phenomenon of enlarged mitochondria was also reported in *Plin5*-S115A mutant mice [76]. However, in contrast to the previous report, mitochondrial fusion markers such as Mfn1, fission protein 1 (Fis1), and dynamin-related protein 1 (DRP1) were downregulated in *Plin5*-S115A mutant mice, whereas MFN2 levels were unchanged, demonstrating that normal mitochondrial activity is highly regulated by functional PLIN5 [76]. This evidence suggests that PLIN5 plays a pleiotropic role in cells from orchestrating lipid metabolism to regulating inflammation and mitochondrial function.

PLIN5 showed implications in the inflammatory response in different models, and strikingly, LCN2, a well-known acute pro-inflammatory protein and critical regulator of lipid uptake, has been shown to act as a regulator of PLIN5 activity in a NASH model based in a methionine- and choline-deficient diet [77]. This report supports that LCN2 impacts PLIN5 expression in a PPAR-independent manner, since its stimulation with an PPAR-γ agonist failed to suppress its expression [77].

Furthermore, studies on *Plin5*^−/−^ mice have revealed a close local connection of mitochondria and LDs [27,78]. In the respective model, the oxidative capacity of mitochondria in cardiac muscle was impaired due to changes in lipid composition of the membrane and its polarization [78]. Super-resolution microscopy-based approaches in skeletal muscle have shown that under basal conditions there are three pools of PLIN5, namely in the cytosol, in close proximity of mitochondria, and surrounding LDs in the mitochondria–LD interface [79]. Moreover, it was observed that increased FA levels led to an augment of PLIN5 in the mitochondria–LD interface pool, suggesting that this relocation dynamic of PLIN5 might facilitate flux of FA from LDs to mitochondria [79,80]. All these data demonstrate the fine-tuned regulation of PLIN5 in lipotoxicity, inflammation response, and mitochondrial dynamics, processes that are all closely related in the development of NAFLD. The major biological processes in which PLIN5 participates are illustrated in Figure 3.

## 6. Perilipins in Cancer: The Unrevealed Role of PLIN5

Liver cancer represents the sixth most common type of cancer and the third highest cause of cancer death in 2020 according to International Agency for Research on Cancer (https://gco.iarc.fr/, last accessed 7 May 2021). Primary liver cancer includes HCC, cholangiocarcinoma, and other rarer types. Several risk factors are associated with the development of cancer such as chronic hepatitis B virus (HBV) and hepatitis C virus (HCV) infections, alcohol intake, obesity, type 2 diabetes, and NAFLD [81,82].

The tumor microenvironment (TME) is a complex net involving cancer cells, immune cells (B cells, T cells, and NK cells), and stromal cells (blood and lymphatic endothelial cells and cancer associated fibroblasts). Moreover, diverse molecular components including chemokines, cytokines, extracellular matrix compounds, and soluble immunosuppressive molecules act coordinately to maintain chronic inflammation and immune suppression leading to tumorigenesis [83]. In HCC, a heterogenous cell population consisting of progenitor cells, hepatocytes, and adult stems cells present significant transcriptional changes, thereby leading to chronic inflammation, giving place to cancer stem cells and promoting aberrant differentiation which ultimately leads to tumor progression and metastasis [13]. In order to identify these cell populations with stem cells properties, some surface markers such as epithelial cell adhesion molecule (EpCAM), cluster of differentiation (CD)44, CD90, and CD33 have been proposed [13].

Nowadays, there is a considerable knowledge on how metabolic changes in lipid homeostasis contribute to aberrant cell growth and cancer progression. For instance, the accumulation of saturated or unsaturated FA levels triggers cellular stress and altered cascades of signaling of lipid metabolism [14,83]. Proper homeostasis of intracellular FA is required for biosynthesis of biological membrane lipids, signaling molecules, and post-translational modifications of proteins. Impairment in this balance is taken advantage of by cancer cells which use it to benefit their proliferation rate by acquiring FA of exogenous sources (FA accumulation sites) or by endogenous synthesis. Two of the mechanisms altered in order to guarantee cancer cell survival are de novo lipogenesis and β-oxidation processes, suggesting that alterations in lipid metabolism can act as a general hallmark of cancer [13,14,15]. Indeed, samples of HCC revealed the upregulation of the transcriptional factor sterol regulatory element-binding factor 1 (SREBF1) whose expression was correlated with high mortality [84]. A recent meta-analysis study revealed a positive correlation between the overexpression of SREBP1, fatty acid synthase (FASN), and stearoyl-CoA desaturase-1 (SCD1) in HCC, while a poor prognosis was associated with increased levels of these markers [85]. The above findings suggest that activation of the lipogenic pathways is fundamental in the pathogenesis of HCC.

Advanced technologies have been useful to detect novel targets and biomarkers for estimating prognosis in cancer. Perilipins have also recently come into focus of such studies. For example, RNA-seq and microarray analysis revealed that PLIN1 can participate in breast cancer development, while patients with upregulated levels of *PLIN1* mRNA present lower overall survival rates [86]. Immunohistochemical validation of these findings in human specimens confirmed these findings as well as in MCF-7 and MDA-MB-231 cell lines, where overexpression of PLIN1 suppressed cellular proliferation, migration and invasion, possibly indicating a role of PLIN1 as tumor suppressor gene [86]. A similar approach performed in clear cell renal cell carcinoma (ccRCC) revealed that PLIN2 upregulation is associated with a good disease prognosis, while *PLIN2* knockdown in human kidney carcinoma A-498 cells favors proliferation, migration, and invasion of ccRCC cells [87]. In prostate cancer, PLIN3 upregulation has been associated with poor prognosis in patients [88]. Moreover, elevated PLIN3 expression has recently been related to poor prognosis of lung adenocarcinoma, whereas ablation of PLIN3 inhibited the invasion ability of lung adenocarcinoma cells [89].

Information contained in the Human Protein Atlas indicates PLIN5 level alterations in some types of cancer (http://www.proteinatlas.org/, last accessed 24 March 2021). A first study about participation of perilipins in neoplastic steatogenesis was reported by Straub and colleagues. They suggested that PLIN1, PLIN2, and PLIN3 may be involved in cancers such as hepatocellular adenoma and carcinoma, sebaceous adenoma and carcinoma, and lipoma tumors, while PLIN5 was excluded as not sufficient proof was given [90].

Little is understood so far about how PLIN5 is involved in the development or progress of cancer, but a novel transcriptome analysis of lung cancer, predicted that higher expression of the guanine nucleotide-binding protein α-inhibiting activity polypeptide 3 (GNAI3) and the lncRNA AC087521.1 was associated with a short survival time, while the NHL repeat-containing protein 2 (NHLRC2) and PLIN5 correlated with longer survival times [91].

Regarding HCC, recent reports have shown increased expression of PLIN5 in HCC, ccRCC, and lipo-, rhabdo- and leiomyosarcoma by immunohistochemical analysis [92]. More recently, perilipins were reported as markers of liposarcoma and liposarcoma subtypes. Using a set of 245 samples and analyzing tissue microarray and immunohistochemistry expression, the results showed that PLIN2, PLIN3, and PLIN5 were commonly expressed in liposarcomas, rhabdomyosarcomas, leiomyosarcomas, dermatofibrosarcoma protuberans, undifferentiated sarcomas, fibrosarcomas, Ewing’s sarcomas, and epithelioid sarcomas [93]. Opposite to these reports, *Plin5* expression is downregulated by promoter methylation in ovarian cancer. Furthermore, an in vivo model, revealed the possible role of PLIN5 as a tumor repressor gene that abrogates cell proliferation, migration, and invasion, thus promoting apoptosis [94]. These findings underpin the pleiotropic activity of PLIN5 in cancer.

Information about PLIN5 implication in HCC is still limited. It was first shown in 2019 that PLIN5 expression is upregulated in HCC tumoral areas in murine HCC models and human HCC biopsy specimens as assessed by Western blot analysis and quantitative real time PCR [95]. RNA-seq and splicing factor (SF) data of 390 HCC cases deposited in the Cancer Genome Atlas (TCGA) data portal (https://tcgadata.nci.nih.gov/tcga/, last accessed 24 March 2021) indicate that PLIN5-46808-AT was found more common in primary sites of HCC with metastasis than primary HCC [96]. It is evident that more studies elucidating HCC development are necessary to clarify the actual function of PLIN5 in HCC.

## 7. Conclusions

This review discusses PLIN5 as a key regulator of pathways involved in pathogenesis of NAFLD and cancer. Since the discovery of PLIN5 in 2006, its fundamental role in lipid metabolism has been extensively studied. PLIN5 regulates lipid metabolism, reduces oxidative stress and ER stress, and promotes autophagy, making this member of the perilipin protein family an interesting therapeutic target. However, the precise mechanisms by which PLIN5 protects the liver from injury are still poorly understood. Depending on the tissue studied, effects of *Plin5* depletion can be contradictory, possibly reflecting its fine-tuned tissue-specific regulation and activity. Broadening the network of PLIN5 targets and cofactors in distinct stages of NAFLD-HCC pathogenesis will help to characterize more specific roles of PLIN5, potentially allowing the use of this protein as a tool for diagnosis or therapeutic targeting. Hot topics are currently the mitochondrial functions affected by PLIN5 and its participation in HCC initiation and progression.

## Figures and Tables

**Figure 1 ijms-22-05284-f001:**
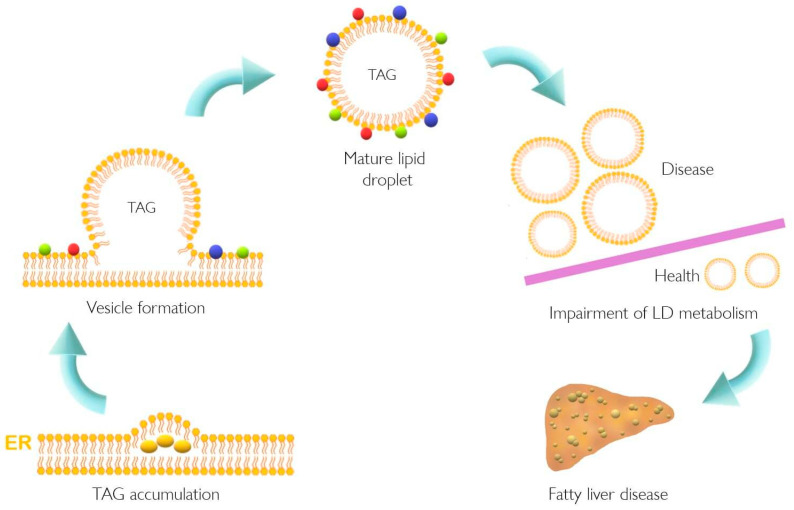
Lipid droplet formation and excessive accumulation leads to hepatic pathological conditions. TAG accumulation in the ER membrane, accompanied by the action of different proteins, leads to LD formation. This excessive accumulation results in impairment in storage of LD in several tissues resulting in pathological conditions such as non-alcoholic fatty liver disease. Colorful spheres around LD represent perilipins, Rab proteins, and Arf-COPI proteins. ER, endoplasmic reticulum; LD, lipid droplet; TAG, triacylglyceride.

**Figure 2 ijms-22-05284-f002:**
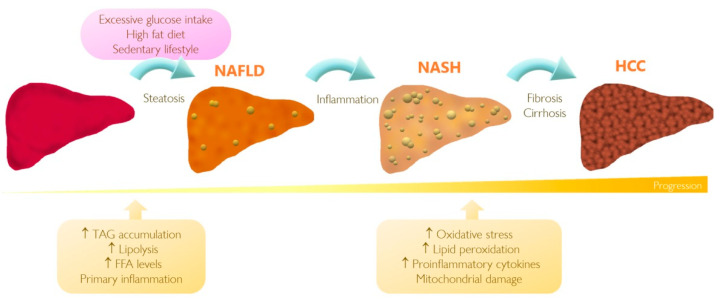
Progression of non-alcoholic fatty liver disease (NAFLD) to hepatocellular carcinoma (HCC). A passive lifestyle accompanied with high caloric diet intake can result in impairment of hepatic lipid metabolism and consequently progression of fatty liver disease. Although the stages of this pathology are not fully characterized yet, the impairment affects expression of genes regulating oxidative stress, lipid peroxidation, mitochondrial damage, and inflammation. When the corresponding noxae persists, simple hepatic steatosis can progress to the development of non-alcoholic steatohepatitis (NASH) and possibly HCC. FFA, free fatty acid; TAG, triacylglyceride.

**Figure 3 ijms-22-05284-f003:**
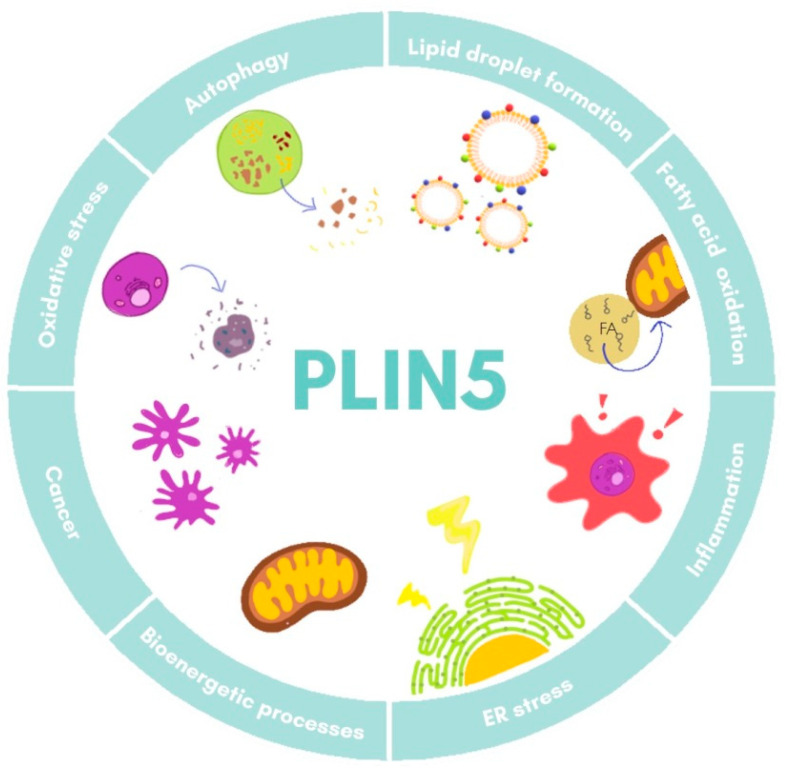
Cellular functions of PLIN5. Major processes in which PLIN5 has been reported to participate are shown. ER, endoplasmic reticulum.

**Table 1 ijms-22-05284-t001:** Phenotypes of *Plin5* transgenic mouse models.

Model	Tissue	Findings	References
*Plin5*^−/−^ mice	Heart	No LD formation in fed and fasting state ↓ TAG content in fed and fasting state	[59]
	Soleus muscle	↓ TAG content	
	Liver	↓ TAG content in fed state ↑ TAG content in fasting state	
	BAT	↓ TAG content in fed and fasting state	
	WAT	↓ TAG content in fasting mice	
*Plin5*^−/−^ mice	Whole body	↑ Carbohydrate oxidation	[20]
	Muscle	↑ Skeletal muscle insulin resistance	
	Liver	Improvement of insulin sensitivity	
*Plin5*^−/−^ mice	Heart	↓ Cardiac LD formation ↑ Cardiac FA oxidation	[34]
*Plin5*^−/−^ mice	Liver	↓ Hepatic TAG content ↑ Lipolysis ↑ Mitochondrial proliferation ↑ Mitochondrial oxidative capacity ↑ Expression of pro-inflammatory genes under an HFD ↑ Expression of ER stress-related genes ↑ Lipid peroxidation	[23]
*Plin5*^−/−^ mice	Liver	Under HFD vs. CtrD: ↓ Fat scoring ↓ Ballooning of hepatocytes ↓ Levels of liver damage enzymes ALP, ALT, AST ↓ Bilirubin levels ↓ Cholesterol levels ↑ Levels of mitochondrial structure and trafficking markers ↑ Lipogenesis ↓ Inflammatory markers ↓ Levels of arachidonic acid	[27]
MKO mice	Skeletal muscle	↑ Fat mass ↓ Respiratory exchange ratio ↑ FA oxidation under HFD ↑ Oxidative stress ↑ TAG content ↓ Pro-inflammatory markers	[25]
	Heart	↓ TAG content No ER stress, inflammation and oxidative stress	
Hepatocyte-specific *Plin5*^−/−^ mice	Liver	↓ FA consumption ↓ FA oxidation ↓ TAG secretion ↓ Lipid peroxidation and oxidative stress Insulin resistance and enhancement under an HFD Glucose intolerance under HFD TAG accumulation under HFD	[21]
CM-*Plin5* mice	Heart	↑ Accumulation of TAGs ↑ TAG hydrolytic activities: ↑ ATGL and CGI-58 protein levels Moderately reduced FA oxidizing gene expression levels	[60]
MCK-*Plin5* mice	Skeletal muscle	↑ LD formation ↓ Body weight compared to non-transgenic littermates under control and HFD diet ↑ Expression of ER stress markers	[61]
	Heart	↑ TAG content	
	Diaphragm	↑ TAG content	
	EDL	↑ TAG content	
	Gastrocnemius	↑ TAG content	
	Liver	↓ Cholesterol levels in an HFD compared to control diet littermates ↓ Lipid uptake ↓ Inflammatory markers	

ALP, alkaline phosphatase; ALT, alanine aminotransferase; AST, aspartate aminotransferase; BAT, brown adipose tissue; CtrD, control diet; CM-*Plin5*, cardiac muscle-specific overexpression of *Plin5*; EDL, *extensor digitorum longus* (muscle); ER, endoplasmic reticulum; FA, fatty acid; HFD, high fat diet; LD, lipid droplet; MCK-*Plin5*, skeletal muscle-specific overexpression of *Plin5*; MKO, muscle-specific *Plin5* knockout; *Plin5*^−/−^, *Plin5* deficient; TAG, triacylglyceride; WAT, white adipose tissue.

## Data Availability

This review includes no original data.

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
