# Peer review of "Understanding the Role of Perilipin 5 in Non-Alcoholic Fatty Liver Disease and Its Role in Hepatocellular Carcinoma: A Review of Novel Insights"

_ijms, 2021, doi:10.3390/ijms22105284_

Round 1

Reviewer 1 Report

The authors should be congratulated on choosing a relevant topic with new insights regarding the development of NAFLD to cancer. However, I recommend that the manuscript undergoes language editing in order to become more understandable to readership outside of the fields of NAFLD and perilipin 5.

General comments:

Authors use numerous abbreviations to describe the different pathways in the manuscript. Therefore, a list of abbreviation would be appreciated.

The writing of PLIN5 highly varies within the manuscript (PLIN5, PLIN5, Plin5, Plin5). The author should use consistent abbreviation regarding nomenclature of genes/mRNA and protein for human (PLIN5; PLIN5) and mouse (Plin5, PLIN5).

Specific comments:

Line 32: Define NASH here, not in line 141 as it is used here the first time. Did the author really mean “NASH” or “NAFLD”? If NASH is meant, consider to explain here that NAFLD encompasses a spectrum of liver diseases including simple steatosis and NASH as the progressed form of NAFLD.

Lines 67-77: The two paragraphs actually describe the same. Consider to summarize these parts.

Line 69: Sentence “In this line…“ Did the authors mean “in this regard“ or “Accordingly“?

Lines 92-93: “by PPARα agonists” is mentioned double.

Lines 96-97: “activating transcription factor 1 (ATF1), activating transcription factor 3 (ATF3) and activating transcription factor 4 (ATF4)”; better: activating transcription factor (ATF) 1, ATF3 and ATF4.

Line 121: “has been proposed…”            

Line 139: acids; rellocation

Lines 175; 413; 414; 418: Describe cell lines shortly for better understanding for the Reader, e.g. the murine hepatocyte cell line AML-12

Lines 238-244: Why is it important that PLIN5 attenuates HSC activation? Consider to mention the role of HSCs as main driver of liver fibrosis due to deposition of collagen and other ECM protein.

Lines 302-313: Consider to shorten this paragraph as many abbreviations are used to describe genes from PPARa pathway, which are not described in liver nor in other cells/tissue in the following.

Line 348: to regulate or regulating

Lines 374; 424; 452: remove links to references.

Lines 388-389: see comment for lines 96-97.

Reviewer 2 Report

Understanding the Role of Perilipin 5 in Non-Alcoholic Fatty Liver 

Disease and its Recent Appearance in Cancer

ijms-1176402

Report

Thank you for asking me to review the above-titled manuscript. The article addresses an important topic, and the authors have covered the current literature. However, there are a few issues that need the authors' revision.

  1. Title: "..Its recent appearance in cancer." Not clear and non-specific- may be changed to "its role in HCC development."
  2. Abstract: (1) did not explain why perilipin 5 was selected for this review; why it should be studied concerning NAFLD, and HCC (2) Lines 10-20 represent a long introduction, could be reduced. (3) The authors should mention pro-inflammatory cytokines, mitochondrial damage, ER stress, insulin resistance etc, to justify this connection.
  3. Page 3, line 124- What are the pro-inflammatory cytokines involved [ref, 45]. Are there other papers exploring this area?
  4. The summary in Table 1 can be used to make Figure 2 more detailed, explaining the pathogenesis at cellular and molecular levels in more details.
  5. Page 12: conclusions lines 457-466: could address key issues of what we know and areas that need further research or we do not know.
  6. References:

This reference could be added

Atorvastatin reduces lipid accumulation in the liver by activating protein kinase A-mediated phosphorylation of perilipin 5.Gao X, Nan Y, Zhao Y, Yuan Y, Ren B, Sun C, Cao K, Yu M, Feng X, Ye J.Biochim Biophys Acta Mol Cell Biol Lipids. 2017 Dec;1862(12):1512-1519. doi: 10.1016/j.bbalip.2017.09.007. Epub 2017 Sep 13.

This reference is not correctly written.

Perilipin-5 is regulated by statins and controls triglyceride contents in the hepatocyte.Langhi C, Marquart TJ, Allen RM, Baldán A.J Hepatol. 2014 Aug;61(2):358-65. doi: 10.1016/j.jhep.2014.04.009. Epub 2014 Apr 21.

Round 2

Reviewer 2 Report

The authors have addressed the points raised by the reviewer.